# Ketamine as Add-On Treatment in Psychotic Treatment-Resistant Depression

**DOI:** 10.3390/brainsci13010142

**Published:** 2023-01-13

**Authors:** Maria Gałuszko-Węgielnik, Zuzanna Chmielewska, Katarzyna Jakuszkowiak-Wojten, Mariusz S. Wiglusz, Wiesław J. Cubała

**Affiliations:** Department of Psychiatry, Faculty of Medicine, Medical University of Gdańsk, Smoluchowskiego 17, 80-214 Gdańsk, Poland

**Keywords:** ketamine, esketamine, treatment-resistant depression, psychotic depression

## Abstract

Psychotic treatment-resistant depression is a complex and challenging manifestation of mood disorders in the clinical setting. Psychotic depression is a subtype of major depressive disorder characterized by mood-consistent hallucinations and/or delusions. Psychotic depression is often underdiagnosed and undertreated. Ketamine appears to have rapid and potent antidepressant effects in clinical studies, and the Federal Drug Agency approved the use of ketamine enantiomer esketamine-nasal spray for treatment-resistant depression pharmacotherapy in 2019. This study aimed to assess the usage of ketamine for major depressive disorder with psychotic features as an add-on treatment to the standard of care. Here we present four inpatients suffering from treatment-resistant depression with psychotic features, including one with severe suicidal crisis, all treated with 0.5 mg/kg intravenous infusion of ketamine. Subsequent monitoring revealed no exacerbation of psychotic symptoms in short and long-term observation, while stable remission was observed in all cases with imminent antisuicidal effect. Results suggest ketamine may benefit individuals with treatment-resistant depression with psychotic features.

## 1. Introduction

Major depressive disorder (MDD) with psychotic features is associated with a less favorable prognosis and outcome as compared to non-psychotic MDD. It is associated with more burdensome symptomatology, course, and a high relapse rate. The exact epidemiology of psychotic depression is unclear as it is frequently misdiagnosed. However, the lifetime prevalence of MDD with psychotic features in the general population is reported from 0.35% to 1% [1]. The psychotic trait may be transmissible irrespective of other characteristics, such as instability of mood or thinking. Consequently, a relationship between a psychosis characteristic and dysregulated mood may result in a psychotic mood disorder profile [2].

As known treatments for MDD with psychotic features are approved, therapies are based upon clinical treatment guidelines supported by registry data and prioritize combined treatments with antidepressants (ADT) and antipsychotics or electroconvulsive therapy ECT [3,4]. However, a proportion of patients with MDD with psychotic features present psychotic treatment-resistant depression (TRD) and do not respond to recommended therapeutic interventions.

The observation that the dissociative anesthetic drug ketamine produces a rapid antidepressant effect in treatment-resistant patients with MDD, along with extensive research with ketamine racemic mixture as well as it is (R)-ketamine (arketamine) and (S)-ketamine (esketamine) enantiomers led to the clinical development and regulatory approval for esketamine-nasal spray (ESK-NS) as an add-on treatment to oral ADT in TRD in acute management and relapse prevention [5]. Consequently, ESK-NS was approved in the acute management of MDD, depressive symptoms in adults with MDD with acute suicidal ideation or behavior (MDSI), and major depressive disorder with a psychiatric emergency (MDD-PE) as an add-on treatment to the standard of care (SOC). With the onset of rapid-acting ADT (RAAD), a high proportion of patients who have been considered treatment-resistant and difficult to treat may benefit from a novel pharmacological approach for mood disorders.

Ketamine and its enantiomers exhibit a favorable safety and tolerability profile. The commonly reported adverse drug reactions include psychotomimetic phenomena that are dose-related and self-limiting in time with no sequelae and symptomatology abatement at the time of treatment [6]. Industry clinical trials and the majority of academic research registries exclude patients with psychotic features, as there is a concern about psychosis exacerbation due to ketamine. On the contrary, few data on MDD with psychotic features indicate a good safety and tolerability profile with no psychotic exacerbation. MDD with psychotic features represents an extreme manifestation of a major depression episode (MDE). Major depressive disorder, with the psychotic feature itself or vulnerability to psychosis is a life-threatening condition, but being complicated with treatment resistance to SOC treatment, the options are limited [7,8,9,10]. 

There is evidence supporting the pro-cognitive effects of ketamine, which imply that ketamine may decrease cognitive performance during infusion or within a few hours following infusion but improve cognition after infusion. Malhotra et al. [11] demonstrated a worsening of cognitive symptoms in patients with schizophrenia following ketamine infusion; the effects were only noticed during the infusion and disappeared 90–120 min later. Therefore, patients with psychotic depression and cognitive symptoms may benefit from exposure to ketamine for a limited period of time [12,13].

There is significant evidence that an increase in synaptic neuroplasticity is essential for the antidepressant effect of ketamine. In addition to alleviating symptoms and boosting neurobiological learning processes, this approach seems to aid patients by promoting early involvement in cognitive-behavioral therapy (CBT). This therapeutic combination of ketamine and CBT may result in better long-term results while reducing the requirement for continued ketamine exposure [14,15]. In addition, early data suggests that ketamine might enhance negative self-schemas, which is one of the primary therapy goals of CBT [16].

Both academic research and clinical trials indicate that ketamine, with its enantiomers use, does not precipitate psychotic features; however, there is some limited evidence. There is also limited evidence for efficacy with good safety and tolerability in these patients, with scarce reports available to date. 

As literature reports on ketamine use in psychotic TRD are scarce, the reports on the intervention outcome with regard to efficacy and, above all, safety and tolerability profile are of vital interest to practitioners.

This paper reports on four patients presenting with psychotic TRD demonstrating data on the safety and tolerability of short-term ketamine use as an add-on treatment to SOC. The study was carried out in accordance with the latest version of the Declaration of Helsinki. For each participant, written consent was obtained after the procedures had been fully explained. The study protocol was approved by the Independent Bioethics Committee for Scientific Research at the Medical University of Gdańsk, Poland. The study population comprises MDD subjects treated with ketamine registered in the naturalistic observational protocol of the tertiary reference unit for mood disorders (NCT04226963).

## 2. Materials and Methods

The inclusion criteria were a diagnosis of major depressive disorder (MDD) or bipolar disorder (BD) according to DSM-5 criteria; treatment-resistant depression (TRD) was defined as the patient failing to achieve remission within the 8-week trial of an antidepressant or combination at a therapeutic dose after at least two trials of first-line evidence-based treatments and/or electroconvulsive therapy (ECT).

The exclusion criteria included pregnancy and breastfeeding, hypersensitivity to ketamine, uncontrolled hypertension, and other uncontrolled somatic conditions that, in the opinion of the investigators, may compromise safety.

The structured Montgomery–Asberg depression rating scale interview was used to evaluate the severity of depression symptoms (MADRS-Sigma). Before infusions, on days 1, 8, 14, and 21, as well as one week afterwards, outcome measurements were collected (day 35). Psychometric endpoints included rates of response (50% reduction from baseline MADRS total score) and rates of remission (MADRS 12 at day 21) in addition to the Columbia suicide severity rating scale (C-SSRS), Young mania-rating scale (YMRS), clinician-administered dissociative states scale (CADSS), and brief psychiatric rating scale (BPRS) prior to and during ketamine administration.

All patients described in this paper suffered from MDD from one to ten years. They were admitted to the hospital due to the current MDE with psychotic features. As a result of MDE, one of the patients attempted suicide, the other one suffered from substantial psychomotor retardation, the third one presented severe weight loss, and the fourth one had suicidal thoughts. Table 1 covers further details about patients. 

During hospitalizations, which lasted from 2 to 6 months, all of the patients were treated with multiple ADT. None of them showed any results. In addition, patient number 1 had performed four ECT procedures, which resulted in delirium implicating ECT discontinuation. Table 2 covers further details about patients’ hospitalization. 

Due to the ineffectiveness of SOC treatment and after considering the risks and benefits, all of the patients were administered intravenous ketamine. It is advised that intravenous ketamine be administered at a rate of 0.5 mg/kg over a period of 40 min; dosages may be reduced in situations of intolerability, although effectiveness at lower doses may be diminished. In individuals with TRD, the optimal frequency of intravenous delivery has not been determined [17]. The acute antidepressant effectiveness on day 15 did not vary between twice weekly and thrice weekly intravenous injections (0.5 mg/kg) in multicenter, double-blind research including individuals (N = 67; ages 18–64) with TRD [18].

Following these results, the dosage was 0.5 mg/kg as an intravenous infusion twice a week over 40 min. The treatment duration was four weeks, making a total of 8 ketamine infusions per patient. 

All patients received elements of cognitive-behavioral intervention, including psychoeducation and behavioral activation throughout the hospitalization on a twice-weekly regimen. 

## 3. Results

All of the subjects demonstrated rapid mood improvement with full remission in psychotic symptomatology observed post-ketamine infusion.

Patient 1 received eight ketamine intravenous infusions as an add-on treatment with backbone psychotropic treatment, including olanzapine up to 20 mg/day. During the first infusion, the patient felt nauseated—the symptom was remitted within 20 min, and no other adverse events (AE) were observed across infusions and in follow-up. The subject demonstrated rapid mood improvement with full remission in psychotic symptomatology observed post the first ketamine infusion. On discharge, she received: venlafaxine ER 225 mg, mirtazapine 45 mg, and olanzapine 5 mg as prophylactic treatment as a 1-year follow-up. Full remission is sustained as SOC therapy and shows the full recovery, almost to premorbid functional status.

Patient 2, after the 4th infusion, regained psychomotor efficiency. The psychotic symptoms were asymptomatic on discharge and at a 2-year-long follow-up. During the first, second, and third ketamine infusions, he reported mild dizziness remitting within 10 min. No other AE appeared. On discharge, his medication was: sertraline 200 mg and quetiapine 200 mg.

Patient 3 neglected any adverse effects during eight ketamine intravenous infusions. The subject’s state improved after the 3rd infusion, as she no longer demonstrated nihilistic delusions. After the 6th infusion, MDE was asymptomatic. On discharge, she received venlafaxine ER 300 mg, mirtazapine 45 mg, and pregabalin 150 mg.

Patient 4 presented rapid improvement after the 2nd infusion—neither psychotic symptomatology nor suicidal thoughts were observed. After the first and second infusions, she reported mild dizziness remitting within 20 min. No other AE was observed. On discharge, she received venlafaxine ER 375 mg, mirtazapine 45 mg, and lurasidone 74 mg.

Table 3 covers the observed adverse effects of ketamine infusions.

## 4. Discussion

This report supports evidence for good ketamine safety and tolerance in TRD with psychotic symptoms, including one case with suicidality. In line with limited previous reports, no psychotic exacerbation was observed in short and long-term observation [19,20]. To our best knowledge, this is the second report of MDD with psychotic features and suicidality.

Psychotic TRD represents the most severe presentation of MDD, characterized by a high mortality rate and unfavorable prognosis [1]. The report on the multiple intravenous infusions of ketamine for TRD referred to good efficacy and safety profiles in four subjects with TRD with psychotic symptoms as an add-on treatment to SOC, and observations corroborated with safety and tolerability profiles in the domain of dissociation being similar to non-psychotic TRD and characterizing a dose-related self-limiting and transient pattern [7,8,21,22,23].

Psychotic symptoms were found in a median of 19% of studies that included both inpatients and outpatients or only outpatients, compared to 42% of studies that included only inpatients. Psychotic symptoms are most prevalent in patients with severe depression and are related to poorer treatment results and increased relapse rates [1]. In addition, up to sixty percent of individuals with schizophrenia exhibit negative symptoms that coincide with those of depression [24]. Patients with depression and those with negative symptoms of schizophrenia do not respond favorably to antidepressants and antipsychotics. Anti-anhedonic effects of subanesthetic dosages of ketamine have been observed without a deleterious impact on the long-term psychotic symptomatology of schizophrenia patients. In a new meta-analysis, the authors synthesize the research on the effects of ketamine for depression in patients with a history of psychosis or current psychotic symptoms, as well as trials evaluating ketamine as a treatment for negative symptoms in schizophrenic patients [7]. Analyzed were nine papers of pilot trials and case reports involving a total of 41 individuals. As side effects were mild and self-limiting, these investigations imply that short-term ketamine treatment for depression and even negative symptoms in patients with a history of psychosis or current psychotic characteristics can be both safe and beneficial. 

A recent post-hoc analysis investigates whether a lifetime history of psychosis influences patients’ response to a single infusion of ketamine in clinical trials for depression. The authors merged the results of three randomized, placebo-controlled crossover trials with depressed patients. Of the 69 patients evaluated, two were diagnosed with MDD with psychotic features and ten with bipolar disorder with psychotic features in the past. This study reveals that a single infusion of ketamine produces antidepressant effects in people with a history of psychosis without generating psychotic symptoms [21].

Two patients were described as the first to use ketamine as an antidepressant for patients with present psychotic symptoms. One was diagnosed with MDD and the other with schizoaffective disorder; both were suffering from MDE with psychotic symptoms. Infusions of ketamine were safe and well tolerated, with rapid cessation of psychotic symptoms [8].

Ajub and Lacerda [22], described the efficacy of esketamine in four patients with psychotic characteristics and severe depression, following the description of these two individuals. Two patients were diagnosed with major depressive disorder with psychotic symptoms, one with bipolar depressive disorder with mixed features, and one with schizoaffective disorder, depressive type. Three patients exhibited considerable improvement or complete remission of both depressed and psychotic symptoms 24 h after esketamine injection and after two and four weeks of follow-up examination. The administration of esketamine to these four patients did not aggravate their psychotic symptoms.

In adolescents with mixed TRD and psychotic traits, repeated infusions of ketamine over 3 and 8 weeks decreased depressive symptoms, suicidal thoughts, and psychotic symptoms [25,26].

In the majority of investigations, dissociative effects of ketamine were recorded; most were mild and temporary, occurring during the initial dose and diminishing over time, with complete resolution occurring within 60 to 80 min post-infusion [21,22,25].

The research suggests that there was no psychotic exacerbation among the participants in the included studies on this topic. In several cases, comorbid psychotic symptoms improved or completely disappeared after the administration of ketamine or esketamine for depression, consistent with the theoretical notions underlying the treatment of TRD with the novel pharmacological approaches [27]. 

Changes in diagnostic criteria make it challenging to understand psychotic depression studies conducted throughout the years. The clinical and conceptual impacts of a survey are contingent on whether psychosis is characterized by delusions and/or hallucinations, severity, impairment, or melancholia, regardless of whether MDEs were unipolar or bipolar and whether the control subjects comprised of MDD of comparable severity without psychosis, psychosis without depression, or melancholic versus non-melancholic depression.

Although psychosis may not always precede every future depressive episode, it operates as an autonomous component of depression that fundamentally modifies mood disorders, distinguishing it from other kinds of depression [28,29,30].

Inasmuch as psychosis, rather than the degree of depression, tends to modify the course and treatment response of mood disorders, identifying psychosis in depressed patients, especially those who do not react as predicted to antidepressant medication, is a critical therapeutic endeavor. Psychosis may be less evident in clinical practice than in research studies. This paper supports evidence for the distinct phenomenology of psychosis and dissociation that pose separate diagnostic entities. The idea of downstaging, which influences the course and prognosis of mood disorders, is consistent with psychotic TRD as determined by the disease’s development stage and the ketamine effect found in the studied instances [31].

Since psychosis is characterized by perceptual and cognitive abnormalities that often emerge prior to the beginning of the illness [32], biomarkers of brain functions in connection to information processing are especially intriguing for elucidating its origin. Due to their remarkable temporal resolution, neurophysiological methods, particularly electroencephalography (EEG) and event-related potentials (also known as evoked potentials), which are changes in the EEG induced by stimuli, are promising for the study of perception and cognition in vivo. In addition to abnormalities found in patients, neurophysiological indicators of genetic and clinical risk for psychosis might enhance our mechanistic knowledge and perhaps improve therapeutic management. However, imaging methods, such as neurophysiological indicators of genetic risk (endophenotypes), are also essential for elucidating disease causes. In case-control studies, it is important to include possible confounders, notably antipsychotic drugs. The great majority of neurophysiological investigations included antipsychotic-treated subjects. Although medication-free first-episode psychosis patients demonstrate comparable neurophysiological abnormalities to chronic patients, these deficits are often considered to be less severe [33]. The combined use of neurophysiological indicators and genetics has the potential to shed light on the genesis of psychosis and assist the development of novel medications, whilst research on high-risk individuals might expedite psychosis patients’ access to psychological and medical therapy.

There are relatively little biomarker data for TRD patients exposed to ketamine and its enantiomers. Review findings show that ketamine may provide an anti-inflammatory impact and reduce at least one inflammatory marker when administered. Neuroimaging studies have indicated that the cingulate cortex is the primary site of ketamine’s effect. The bulk of blood-based, neuroimaging, and neurophysiological studies reviewed demonstrate that ketamine induces normalization of major depressive disorder aetiology through synaptic plasticity and functional connectivity. Currently, no biomarker/biosignature has been well validated for clinical use, although some seem intriguing [34]. 

This report has several limitations. Firstly, it refers to four cases, presenting substantial variability regarding the severity stage and symptomatology of the disease. Moreover, the data reported come out of the naturalistic registry, and the risk of selection bias in post hoc registry data exploration exists. Thus, no causative effect must be drawn as the conclusion, the generalization of the observation is not possible and the tendency for overinterpretation must be strongly restricted. 

In the case of certain food patterns, drug use disorders (such as nicotine dependence), and physical activity levels, it is recognized that they may create both pharmacokinetic and pharmacodynamic consequences that confuse our results. All patients reported above having a healthy diet were nonsmokers and lacked a drug use issue. Their degree of activity was minimal.

Due to the variability of cases presented it can not be replicated. Besides, the follow-up ranges from 2 months to 2 years, with an extensive longitudinal observation missing. Thus, this report is hypothesis generating in value. 

Our results support the notion that a history of psychosis should not exclude people from receiving ketamine in a therapeutic context. However, such patients may be at risk of experiencing more significant short-term dissociative symptoms. 

Future studies should investigate how ketamine treatment affects depressed patients with current psychotic features, given these patients may benefit from a trial of ketamine. 

## 5. Conclusions

This paper contributes to the literature supporting the rapid antidepressant effect in TRD and demonstrates the good safety and tolerability profile of intravenous ketamine use as the add-on SOC in psychotic TRD. All cases described correspond with previous literature reports on safety and tolerability, with neither relevant cardiovascular and hemodynamic changes nor psychomimetic exacerbations in the acute treatment phase as well as in the follow-up.

## Figures and Tables

**Table 1 brainsci-13-00142-t001:** Patients’ data and MDD details.

Subject	Gender	Age	Length of MDD (Years)	Number of Previous MDE
Patient 1	female	63	5	1 with full remission
Patient 2	male	35	10	6 with full remission
Patient 3	female	44	1	3 with full remission
Patient 4	female	66	6	3 with full remission

**Table 2 brainsci-13-00142-t002:** Current MDD hospitalization details.

Subject	Reason of Hospitalization	Lines of Failed ADT in the Current Episode in Time	Treatment in the Current Episode
Patient 1	MDE with psychotic features, suicidal attempt	3 in 6 months	sertraline 200 mg, escitalopram 10 mg, duloxetine 90 mg, haloperdol 5 mg, on discharge: venlafaxine ER 225 mg, olanzapine 10 mg
Patient 2	MDE with psychotic features, substantial psychomotor retardation	4 in 4 months	fluoxetine 30 mg, escitalopram 10 mg, venlafaxine ER 225 mg, mirtazapine 45 mg, zuclopethixol 20 mg, on discharge: sertraline 200 mg, quetiapine 200 mg
Patient 3	MDE with psychotic features, severe weight loss	2 in 5 months	fluoxetine 60 mg, olanzapine 20 mg, amitriptyline 150 mg, on discharge: venlafaxine ER 337.5 mg, mirtazapine 45 mg, pregabalin 150 mg
Patient 4	MDE with psychotic features	2 in 2 months	escitalopram 15 mg, sertraline 200 mg, olanzapine 5 mg, on discharge: venlafaxine ER 375 mg, mirtazapine 45 mg, lurasidone 74 mg

**Table 3 brainsci-13-00142-t003:** The effects of ketamine infusions.

Subject	Infusion Causing Remission of Symptoms	AE
Patient 1	1st	Nausea during 20’ of 1st infusion
Patient 2	4th	Mild dizziness remitting 10’ during 1st, 2nd, 3rd infusion
Patient 3	4th	No AE reported
Patient 4	2nd	Mild dizziness remitting 20’ during the 1st and 2nd infusion

## Data Availability

Not applicable.

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
