# Peer review of "Ketamine as Add-On Treatment in Psychotic Treatment-Resistant Depression"

_brainsci, 2023, doi:10.3390/brainsci13010142_

Round 1

Reviewer 1 Report

Dear Editor,
I really appreciate the opportunity to review the manuscript brainsci-2164818 entitled:
"Ketamine as add-on treatment in psychotic treatment-resistant depression"

I commend the authors for describing this critical and timely issue. The paper is interesting and well-written; however, I would like to highlight some issues that merit revision:

Since these are therapy-resistant patients, the clinician is often inclined to combine drug treatment with psychotherapeutic treatment, which may go to alter the assessment of the pure efficacy of a ketamine add-on. Please, the authors specify whether psychotherapy was used as an add-on treatment in the four cases presented, and if so, what kind and for how long.

Author Response

Gdańsk, 10th January 2023

The Brain Sciences

Editors-in-Chief

Dear Sirs,

Thank you for the reviewer’s remarks. All are addressed in the revised manuscript. Please, find below the point-to-point discussion presenting the Reviewers’ observations addressed.

Reviewer #1:

I would like to highlight some issues that merit revision:
Since these are therapy-resistant patients, the clinician is often inclined to combine drug treatment with psychotherapeutic treatment, which may go to alter the assessment of the pure efficacy of a ketamine add-on. Please, the authors specify whether psychotherapy was used as an add-on treatment in the four cases presented, and if so, what kind and for how long.

It is a very important issue in the combination of psychotherapy and ketamine treatment. Cognitive behavioral treatment is the most widely researched psychological treatment for depression. There is evidence that an increase in synaptic neuroplasticity is essential for the antidepressant effect of ketamine and early data suggests that ketamine might enhance negative self-schemas, which is one of the primary therapy goals of CBT.

All patients received elements of cognitive-behavioral intervention, including psychoeducation and behavioral activation throughout the hospitalization on twice a week regimen.

We present this information on lines 70-77 and 132-134 of the text.

Reviewer #2:

  1. What were the inclusion and exclusion criteria for the selection of the particular cases?

The inclusion criteria were a diagnosis of major depressive disorder (MDD) or bipolar disorder (BD) according to DSM-5 criteria; treatment-resistant depression (TRD) was defined as the patient failing to achieve remission within the 8-week trial of an antidepressant or combination at a therapeutic dose after at least two trials of first-line evidence-based treatments and/or electroconvulsive therapy (ECT).

The exclusion criteria included pregnancy and breastfeeding, hypersensitivity to ketamine, uncontrolled hypertension, and other uncontrolled somatic conditions that, in the opinion of the investigators, may compromise safety.

We present this information on lines 94-101 of the text.

  1. Given the 4 patients included in the study this would be more appropriately defined as a case report rather than a research article per se.

The selection of the article rather than the case series was designed to emphasize the safety and tolerability of ketamine as an adjunctive therapy for psychotic depression. We intended to discuss the challenging management of psychotic depression in more detail.

  1. What was the rationale for the dose and administration regimen of ketamine? How was the protocol determined?

It is advised that intravenous ketamine be administered at a rate of 0.5 mg/kg over a period of 40 minutes; dosages may be reduced in situations of intolerability, although effectiveness at lower doses may be diminished. In individuals with TRD, the optimal frequency of intravenous delivery has not been determined. The acute antidepressant effectiveness on day 15 did not vary between twice weekly and thrice weekly intravenous injections (0.5 mg/kg) in multicenter, double-blind research including individuals (N=67; ages 18–64) with TRD.

The study design is described in the naturalistic observational protocol of the tertiary reference unit for mood disorders (NCT04226963).

We present this information on lines 90-92 and 122-128 of the text.

  1. Was there any clinical validation of the reported improvement for the patients? 

The clinical validation of changes was performed. Structured Montgomery Asberg Depression Rating Scale structured interview was used to evaluate the severity of depression symptoms (MADRS-Sigma). Before infusions, on days 1, 8, 14, and 21, as well as one week afterwards, outcome measurements were collected (day 35). Psychometric endpoints included rates of response (50% reduction from baseline MADRS total score) and rates of remission (MADRS 12 at day 21) in addition to the Columbia Suicide Severity Rating Scale (C-SSRS), Young Mania Rating Scale (YMRS), Clinician-Administered Dissociative States Scale (CADSS), and Brief Psychiatric Rating Scale (BPRS) prior to and during ketamine administration.

We present this information on lines 102-109 of the text.

  1. A short discussion about potential biomarkers and/or clinical assessments would be interesting in terms of determining psychosis levels and response to treatments. This could be included in the discussion section.

This subject is quite intriguing. There are relatively little data on response biomarkers in TRD patients exposed to ketamine and its enantiomers, as well as psychotic levels. Currently, EEG recordings have a potential function as predictors of psychosis and treatment response. No biomarker or biosignature has been validated for clinical usage of ketamine as of yet.

We present this information on lines 243-269 of the text.

  1. Another interesting point to include in terms of the patients would be potential confounding factors such as smoking, other medications, alcohol consumption and potentially diet and physical activity levels. All these factors could have an effect (synergistically) to the potential for psychotic behavior and as such could confound findings. It would be useful if the authors could provide some information regarding those parameters for their patients.

In the case of certain food patterns, drug use disorders (such as nicotine dependence), and physical activity levels, it is recognized that they may create both pharmacokinetic and pharmacodynamic consequences that confuse our results. All patients reported above having a healthy diet were nonsmokers and lacked a drug use issue. Their degree of activity was minimal.

We present this information on lines 276-280 of the text.

Finally, I would like to express my appreciation for the Reviewers’ observations substantially improving the manuscript quality.

With kind regards,

Maria Gałuszko-Węgielnik M.D., Ph.D.

Department of Psychiatry, Medical University of Gdańsk

Smoluchowskiego 17, 80-214 Gdańsk, Poland

mgaluszko@gumed.edu.pl

Reviewer 2 Report

The manuscript under review is a report of 4 patients on the use of ketamine as an additional treatment for persistent psychosis resistant to typical treatment. It is an interesting manuscript albeit with a very small sample size. 

The reviewer would like to present the following points towards the improvement of the manuscript for consideration by the authors.

1. What were the inclusion and exclusion criteria for the selection of the particular cases?

2. Given the 4 patients included in the study this would be more appropriately defined as a case report rather than a research article per se.

3. What was the rationale for the dose and administration regimen of ketamine? How was the protocol determined?

4. Was there any clinical validation of the reported improvement for the patients? 

5. A short discussion about potential biomarkers and/or clinical assessments would be interesting in terms of determining psychosis levels and response to treatments. This could be included in the discussion section.

6. Another interesting point to include in terms of the patients would be potential confounding factors such as smoking, other medications, alcohol consumption and potentially diet and physical activity levels. All these factors could have an effect (synergistically) to the potential for psychotic behavior and as such could confound findings. It would be useful if the authors could provide some information regarding those parameters for their patients.

Author Response

(The authors gave the same response as above.)

Round 2

Reviewer 2 Report

The authors have reasonably addressed the reviewer's points. Proofreading is suggested.